# Trends and determinants of clustering for non-communicable disease risk factors in women of reproductive age in Nepal

**Barun Kumar Singh[1]\*, Shiva Raj Mishra[2,3], Resham B. Khatri[4]**

**1** Health Nutrition Education and Agriculture Research Development, Saptari, Nepal, **2** Nepal Development Society, Bharatpur, Chitwan, Nepal, **3** Faculty of Medicine and Health, University of Sydney, Sydney, Australia, **4** School of Public Health, University of Queensland, Brisbane, Australia

\* barun272@gmail.com

**Data Availability Statement:** The data used in this study are publicly available from the Demographic and Health Surveys (DHS) Program and can be

## Abstract

### Background

Understanding the clustering of two or more risk factors of non-communicable disease, such as smoking, overweight/obesity, and hypertension, among women of reproductive age could facilitate the design and implementation of strategies for prevention and control measures. This study examined the factors associated with smoking, overweight/obesity, and hypertension among Nepalese women of reproductive age (15–49 years).

### Methods

This study used the Nepal Demographic and Health Surveys (NDHS) 2016 (6,079 women for smoking and overweight/obesity, 6076 for hypertension) and 2022 (6,957 women for overweight/obesity and smoking status and 3,749 women for hypertension) for comparison of trends of NCD risk factors among women aged 15–49 years. Additionally, for each participant, risk factors score (range of 0 to 3) was created by summing individual risk factors. We assessed the determinants of risk factor clustering using multivariable Poisson regression models with robust sandwich variance estimator to calculate adjusted prevalence ratios using NDHS 2022.

### Results

The national prevalence of overweight/obesity increased from 22.2% in 2016 to 29.2% in 2022 among women of reproductive age. In 2022, the prevalence for smoking, overweight/obesity, and hypertension were 3.8%, 29.2%, and 9.6%, respectively. More than one in four women (28.7%) had one NCD risk factor, while 6.5% had two such risk factors. Higher aged women (40–49 years) were more likely to have multiple NCD risk factors than those aged 15–29 years (APR: 3.19; 95% CI: 2.68–3.80). Those in the richest wealth quintile (APR: 1.52; 95% CI: 1.24–1.85), as well as married (APR: 3.02; 95% CI: 2.43–3.76) and widowed/divorced (APR: 2.85; 95% CI: 2.14–3.80) were more likely to have multiple NCD risk factors. Women from Koshi province (APR: 1.74; 95% CI: 1.41–2.15) had more NCD risk factors

accessed via the following link: https://dhsprogram.com/data/available-datasets.cfm.

**Funding:** The author(s) received no specific funding for this work.

**Competing interests:** The authors have declared that no competing interests exist.

than those from the Sudurpaschim province. Working women also had a higher prevalence of NCD risk factors compared to non-working women (APR: 1.23; 95% CI: 1.06–1.43). Additionally, Hill Janajatis (APR: 1.44; 95% CI: 1.21–1.72) and Dalits (APR: 1.42; 95% CI: 1.15–1.75) women were more likely to have NCD risk factors compared to women of Brahmin hill origin.

## Conclusions

Clustering of two or more NCD risk factors was higher among women aged ≥30 years, those who are currently married or widowed/divorced/separated, working women, and individuals from the wealthiest socioeconomic groups. A higher burden of risk factors underscores the importance of targeted public health interventions, particularly among women from advantaged socio-economic groups, those of affluent regions, and in the workplace.

## Background

Non-communicable diseases (NCDs) represent a leading cause of global mortality and disability. NCDs contribute to 41 million deaths each year worldwide; 77% of deaths occur in Low and Middle-Income Countries (LMICs) including Nepal [1]. According to the Nepal Burden of Disease Study 2019, 71% of the disease-induced deaths were attributable to non-communicable diseases (NCDs) [2], while the Housing and Population Census 2021 reported 98,736 deaths due to NCDs in the last twelve months out of 198,463 deaths [3]. Furthermore, evidence suggests the increasing trend for NCDs and their risk factors over the last two decades, for example, hypertension prevalence was increased from 21% to 25% between 2007 and 2019 in Nepal [4], and increased prevalence of the Diabetes Mellitus (DM) was observed from 3.6% in 2013 to 8% in 2019 [4–6].

Nepal's epidemiological transition is amid gradual economic growth and increased rural-to-urban migration [7, 8]. Consequently, most of the population is adopting a sedentary lifestyle, influenced by changes in dietary habits, improved accessibility and preference for unhealthy foods, decreased levels of physical activity, and alarming rates of smoking and alcohol consumption [4, 9–11]. Women face an elevated risk of developing NCDs as they are more prone to experiencing a combination of behavioural and metabolic risk factors [12–15]. Hypertension can affect the normal functioning of the reproductive system of women, leading to menstrual irregularities, diminished fertility, and maternal mortality [16, 17]. Similarly, being overweight and obese can increase the risk of diabetes, cardiovascular diseases (CVDs), cancer, and mortality, emphasizing the urgent need for lifestyle changes [18].

According to the World Health Organization (WHO), 95% of all maternal deaths were occurred in LMICs in 2020 [19], and NCDs in pregnancy are becoming increasingly important in contributing to poor maternal health outcomes. There have been undergoing an obstetric transition, where maternal deaths are shifting from direct causes such as haemorrhage and infection to indirect causes like NCDs [20]. Additionally, women possessing the risk factors for NCDs have negative repercussions on their reproductive health and their offspring, leading to an intergenerational cycle of NCD risk [21, 22]. Such NCD risk factors could be higher among women in Nepal due to a higher rate of undernutrition, overnutrition, increasing childbearing age, chronic stress, and hypertension during pregnancy [20, 23].

Nepal's health system has addressed the growing burden of NCD risk factors through a multifaceted approach, including national policies and regulations like the multisectoral action plans (2016–2020 and 2021–2025) [24, 25] and the Tobacco Products Control and Regulation Act (2010) [26] aligned with WHO Framework Convention on Tobacco Control (FCTC) [27]. The National Health Education, Information and Communication Center (NHEICC) leads health promotion and behaviour change communication initiatives, including tobacco-specific advocacy and awareness programs implemented through multimedia channels, public and private healthcare facilities, media, and volunteers throughout the countries [28]. Health promotion activities targeting modifiable risk factors such as unhealthy diet/nutrition and physical activities are ongoing in primary healthcare settings in Nepal [29–31]. However, the increasing burden of NCDs in Nepal is often not matched with sufficient healthcare response and is mostly scaled up a blanket approach [32]. Most targeted NCD programs are curative, neglecting the importance of preventive and promotive efforts [32, 33].

The screening and management of NCDs have been integrated into primary health services across 77 districts through the Package of Essential Non-Communicable Diseases (PEN) packages [28]. Overall, this package is expected to strengthen NCD risk factor screening, diagnosis, treatment, and referral services at health posts, primary health care centres, and district hospitals for early detection and management of chronic diseases within the community [34]. However, challenges such as a lack of targeted approach for health promotion interventions, scarce healthcare resources, and inconsistent regulation enforcement require ongoing efforts to strengthen the health system's capacity to manage the increasing burden of NCD risk factors [8, 27, 29, 32, 33]. Women in similar contexts are vulnerable to obesity, physical inactivity, and unhealthy behaviours due to gender norms and social determinants [35]. Women's NCD risk has remained under-recognized due to low awareness, screening, and diagnosis compared to men in low-income settings like Nepal [36, 37]. To successfully fill this gap, it is imperative to take a comprehensive and structured approach to assess women's NCD risk factors, which will help in designing and implementing targeted interventions to address the prevention and control of NCDs. Previous studies were unable to provide disaggregated results by sex [5, 8, 38–40], focused on standalone risk factors such as DM [41] and hypertension [42–44], and included specific geographical areas [36, 38, 40, 43] and age groups [45, 46]. This study, therefore, aimed to i.) analyze the prevalence and trends of NCD risk factors between the two rounds of NDHS 2016 and 2022. ii.) investigate the socio-demographic determinants of NCD risk factor clustering among Nepalese women of reproductive age (15–49 years).

## Methods

### Study population and data sources

The trend of the prevalence of NCD risk factors among women of reproductive age were estimated analysing data from the most recent nationally representative NDHSs (2016 and 2022) [23, 47]. The prevalence and determinants of NCD risk factors among women was analysed using data from NDHS 2022. The NDHS 2016 and 2022 report describes the detailed sampling strategy and participants [23, 47]. Briefly, the NDHS 2022 used a two-stage stratified cluster sampling technique. Each province was initially stratified into urban and rural areas, yielding 14 sampling strata. In the first stage, 476 primary sampling units (PSUs) were selected with probability proportional to PSU size and independent selection in each sampling stratum within the sample allocation. Among the 476 PSUs, 248 were from urban areas and 228 from rural areas. In the second stage, a fixed number of 30 households per cluster were selected using an equal probability systematic selection method for a total sample size of 14,280 families. The NDHS 2022 adopted a universally standardized DHS questionnaire and measured

blood pressure with the validated instrument for the second time in the NDHS series. Blood pressure and anthropometric measurements were only obtained from the systematically selected subsample of 14,845 study participants. From the latest NDHS 2022 dataset (n = 14,845 interviewed women), after excluding those women with missing values for height and weight measurements (n = 7,491) and pregnant women, including postnatal mothers up to two months (n = 390), data were extracted for all eligible women (n = 6,957) aged 15–49 years and included in the final analytical sample for overweight/obesity and current smoking status in this study. Blood pressure was measured among 3,970 eligible women, of which 221 were either pregnant or recently delivered, which was excluded from the analysis, resulting in the final model for hypertension and clustered risk factors (n = 3,749). Similarly, the NDHS 2016 dataset was used to compare trends of selected NCD risk factors at both national and provincial levels. The total sample size was 6,457 women aged 15–49 in NDHS 2016. Applying similar inclusion and exclusion criteria for trend analysis, 378 women (292 pregnant and 86 postpartum mothers within 2 months) were excluded, resulting in a 6,079 final analytical sample for smoking and overweight/obesity analysis and 6,076 for hypertension analysis. Details of the 2016 sampling strategy are documented elsewhere [23, 47].

## Outcome variables

The outcome variables for this study considered the three risk factors associated with NCDs: smoking, overweight and obesity, and hypertension. These specific NCD risk factors were extracted from the NDHS 2022 survey dataset. Furthermore, we defined the clustering of these three risk factors and examined their prevalence and the associated factors.

**Current smoking.** Within the NDHS survey, participants' current smoking status was documented as a dichotomous response.

**Overweight and obesity.** Women's weight and height were measured following the DHS VIII biomarker manual [48]. Women who were pregnant during the survey visit or had given birth within the two months prior were excluded from the study. Body mass index (BMI) was calculated by dividing weight in kilograms by the square of height in meters. Women with a BMI between 25.0 and 29.9 kg/m$^2$ were considered overweight, whereas individuals with a BMI greater than or equal to 30.0 kg/m$^2$ were classified as obesity [23]. We merged the overweight and obesity categories into a single binary variable, defining an individual as overweight or obese if their BMI was 25.0 kg/m$^2$ or higher.

**Hypertension.** Blood pressure was measured for all participants aged 15 and above within a subset comprising about one-fourth of the households. Individuals eligible for blood pressure measurements were contacted and provided with instructions on the procedure—those who consented provided blood pressure measurements. An automatic device with separate cuffs was used for blood pressure monitoring. Three blood pressure readings were taken at intervals of approximately 10 minutes. Typically, the average of the second and third readings was used to represent the participants' blood pressure. Hypertension was diagnosed if the average systolic blood pressure reading was 140 mmHg or higher, the average diastolic blood pressure was 90 mmHg or higher, or if individuals were taking antihypertensive medication at the time of the survey [49, 50].

**Risk factors clustering.** The clustering of NCD risk factors was defined as the presence of multiple risk factors in each participant. Assessment of the clustering of risk factors involved counting the risk factors from smoking, overweight/obesity, or hypertension. Thus, their presence was categorized within a range of 0 to 3.

## Explanatory variables

Previous studies conducted in Bangladesh [12] and Nepal [15] on prevalence and determinants of non-communicable disease risk factors among reproductive-aged women and information available in the dataset were employed as a basis for the selection of potential sociodemographic explanatory variables in this study. Some variables, such as caste/ethnicity, education attainment, marital status, and occupational status, were further categorized for this study. For instance, the Government of Nepal has categorized the caste into six broad categories [28]: i) Dalit (Hill and Terai)); ii) Janajati (Indigenous Hill and Terai); iii) Madhesi (non-Dalit Terai caste groups); iv) religious minorities (Muslims); v) upper caste groups (Brahman/Chhetri) vi) Others (Thakuri and Sanyashi). Based on socioeconomic, geographic, and cultural similarities, other studies, and enough representation in the sample size, caste/ethnicities were categorized into seven groups: a) Brahmin Hill b) Chhetri Hill c) Terai caste (merging Terai Brahmin/chhetri hills with other terai caste and others); d) Dalit (merging terai Dalit and hill Dalit); e) Hill Janajati (merging Newar and hill Janajati); f) Terai Janajati; g) Muslim. Similarly, education attainment was further categorized into four groups: a) No education, b) Primary, c) Some secondary, and d) School Leaving Certificate or Higher (merging SLC/SEE and beyond) consistent with another study [51]. Marital status was categorized into never married, married or living together, and widow/separated/divorced, consistent with previous studies [12, 15]. Occupational status was categorized into four groups: a) not working; b) services (merging professional/technical/managerial, clerical, and sales/service); c) agriculture-self employee d) manual (merging skilled manual, unskilled manual, and others) concurrent with previous studies [12, 15]. Other sociodemographic variables considered in the study were respondent's age categories (15–29 yrs, 30–39 yrs, 40–49 yrs), ecological zone (Hill, Mountain, Terai), native language (Nepali, Maithili, Bhojpuri, Others), place of residence (rural, urban), provinces (Koshi, Madhesh, Bagmati, Gandaki, Lumbini, Karnali, Sudurpaschim), wealth index (poorest, poorer, middle, richer and richest) [23]. Socioeconomic status was measured using a household wealth index, a proxy measure of inequality without income, expenditure, and consumption data [52]. According to NDHS 2022, the household wealth index was derived using principal component analysis. Scores were assigned based on easily collectible data on household consumer goods (such as televisions, bicycles, and cars) and household characteristics (housing construction materials, water access, and sanitation facilities). Household wealth quintiles were then computed by assigning a household score to each household member, ranking each person in the household population by their score, and dividing the distribution into five equal categories comprising 20% of the population [52].

## Statistical analysis

The sociodemographic characteristics of the women involved in the study were presented using mean values and weighted percentages. We calculated the crude prevalence of smoking, overweight/obesity, and hypertension by taking the complex survey design and sampling weights into account. One of the objectives was to evaluate the trend of selected non-communicable disease (NCD) risk factors at national and provincial levels over the recent two NDHSs. For this, this paper calculated and compared the proportions of each risk factor at these levels using an independent two-sample (proportion) Z (Wald) test, assuming equal variance in the distribution of those risk factors across the last two surveys [53]. Details about the independent two-sample Z (Wald) test are provided in **S1 Table**. The association between categorical variables was assessed using the chi-square test. Variables with a chi-square test p-value below 0.25 were included in the multivariable Poisson regression models [54, 55]. Modified Poisson regression models with a robust sandwich variance estimate were used to identify

factors associated with NCD risk factors (smoking, overweight/obesity, and hypertension), producing adjusted prevalence ratios (APRs) and corresponding 95% confidence intervals (CIs) [12]. Variable such as ecological zone was excluded after a multicollinearity check revealed a variance inflation factor (VIF) of $\geq 5$. A Poisson regression model was chosen in this study for several reasons. Firstly, this addresses the limitations of existing logistic regression models in cross-sectional studies with binary outcomes which tends to overestimate the association by odds ratio [56–58]. Secondly, a modified Poisson regression model with a robust sandwich variance estimate was used as an alternative to binary logistic regression, considering the prevalence of binary outcome was $\geq 10\%$ [12, 59, 60]. Thirdly, the number of risk factors present with the individual was aggregated, ranging from 0 to 3, to examine the clustering of risk factors. This analysis used the modified Poisson regression model and a robust sandwich variance estimator to adjust the standard errors of regression coefficients [55]. Using robust standard errors can provide valid inference when variance structure is miss-specified due to either over (variance greater than mean) or under dispersion (variance less than mean). Because the mean was heavily weighted by 'zeros' among those with no NCD risk factors (~ 64%), we didn't interpret the mean risk factor and only interpreted the adjusted risk ratio (ARR =, i.e., ratio of count). All estimates are reported as weighted values otherwise indicated. We used STATA 18 software to perform the statistical analysis by deploying two-tailed tests with statistical significance defined as a p-value of less than 0.05. Provided that the NDHS 2022 survey adopted a two-stage stratified cluster sampling technique, the analysis used recommended sample weights [23]. Further, we adjusted the clustering effect using the "svy" command in STATA.

### Ethical considerations

This study used publicly available deidentified secondary data from the nationally representative household survey (NDHS 2016 and 2022). The data sets used for this study were available to apply for on-line on DHS webpage. The first author got permission from MEASURE DHS/ ICF International to download and use the data for this study. Both NDHSs had obtained ethical approval from the Nepal Health Research Council, Nepal, and ICF Macro Institutional Review Board, Maryland, USA.

## Results

### Trends in the prevalence of NCD risk factors

**Fig 1** shows a significant decreasing trend in smoking prevalence among women in all provinces except for Koshi province over the NDHS survey years 2016–2022. The prevalence of overweight and obesity among women rose significantly across all the provinces, excluding Koshi provinces in the same period. Likewise, the prevalence of hypertension also increased significantly in all provinces except Karnali and Madhesh province. The prevalence of current smoking ranged from 2.3% in Madhesh to 14.1% in Karnali in 2016, which significantly declined in 2022, ranging from 0.6% in Madhesh to 9.1% in Karnali. Overweight and obesity rates increased nationally from 22.2% in 2016 to 29.0% in 2022, notably rising in Bagmati up to 44.3% and Gandaki up to 40.3%. Similarly, hypertension prevalence nearly doubled nationally from 5.0% to 9.6% between 2016 and 2022, with significant increases observed in Koshi (6.5% to 13.4%), Bagmati (4.9% to 11.6%), Gandaki (6.5% to 12.0%), Lumbini (3.1% to 6.3%), and Sudurpaschim (3.1% to 6.3%).

### Characteristics of the study participants from NDHS 2022

**Table 1** depicts the sociodemographic characteristics of the study participants (n = 6,957) for current smoking and overweight/obesity, hypertension, and the clustering of NCD risk factors

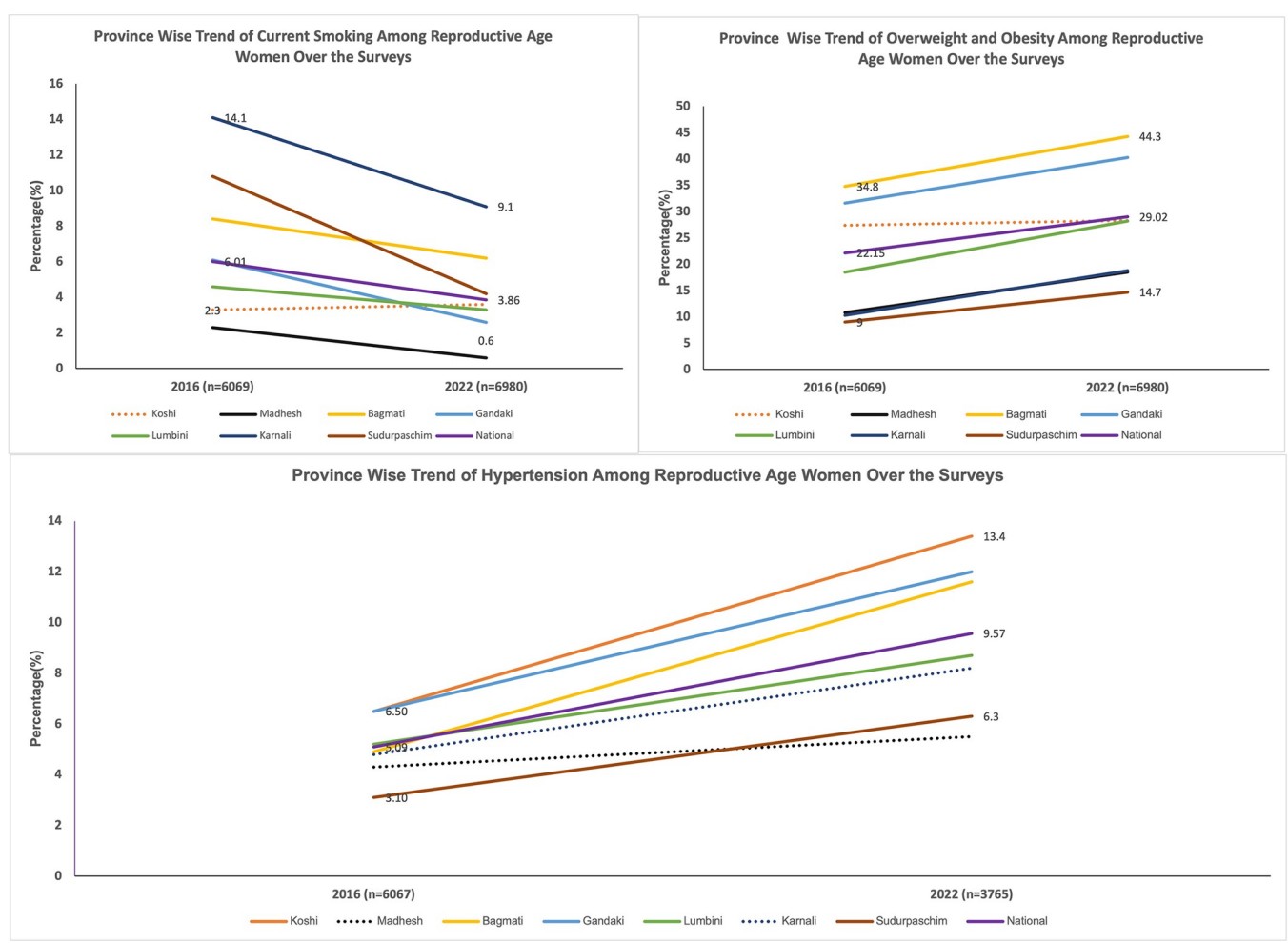

**Fig 1. Trend of NCD risk factors at provincial and national level among women of reproductive age over the surveys.**

(n = 3,749). The mean age of study participants was 30 years (SD = 9.7). Almost two-thirds of them were from urban. Approximately half of them were between 15 and 29 years old. In terms of educational attainment, one-fourth of them did not have any formal education. One-sixth of them had qualifications above the secondary level. Based on the wealth quintile, the richest represented 22.6% of the participants. Approximately half of the participants were engaged in agriculture or self-employment.

**Table 2** depicts the prevalence of risk factors for NCDs among reproductive-aged women. The overall prevalence of smoking was 3.8% among the study participants. Smoking was more prevalent among women from the Mountainous zone (8.4%), Karnali province (9.1%), and Nepalese native speaking (4.6%). Likewise, women with widow/divorced/separated status (9.6%), age group of 40–49 years (9.7%), manual working occupational status (7.1%), no formal education (9.0%), Dalit ethnicity (5.5%) had a relatively higher prevalence of smoking compared to their counterparts (p<0.001). Relative to other risk factors, overweight and obesity were common among the participants (29.2%).

More than four out of ten (44.4%) of the participants between 40–49 years were either overweight or (p<0.001). Similarly, overweight and obesity were more prevalent among women from hills (35.8%), Bagmati province (44.3%), Nepali native speakers (33.4%),

**Table 1. Sociodemographic characteristics of reproductive-aged women (15–49 years) by type of non-communicable disease risk factors (NDHS 2022).**

| Variables | Overweight/obesity and smoking | | Hypertension | |
|---|---|---|---|---|
| | Unweighted Count | Weighted %/Mean (SD) | Unweighted Count | Weighted %/Mean (SD) |
| | n = 6,957 | n = 6,980 | n = 3,749 | n = 3,765 |
| **Age (Years)** | | 30.0 (9.7) | | 30.0 (9.7) |
| **Body Mass Index (Kg/Sq Meter)** | | 23.0 (4.4) | | 23.0 (4.4) |
| **Place of residence** | | | | |
| Urban | 3,751 | 68.6 | 2,026 | 68.9 |
| Rural | 3,206 | 31.4 | 1,723 | 31.1 |
| **Ecological zone** | | | | |
| Mountain | 566 | 5.1 | 295 | 4.9 |
| Hill | 3,146 | 40.7 | 1,712 | 41.1 |
| Terai | 3,245 | 54.2 | 1,742 | 54.0 |
| **Province** | | | | |
| Koshi | 1,054 | 16.9 | 564 | 17.0 |
| Madhesh | 1,126 | 19.4 | 617 | 19.7 |
| Bagmati | 1,008 | 21.4 | 545 | 21.6 |
| Gandaki | 803 | 9.6 | 431 | 9.5 |
| Lumbini | 1,070 | 18.0 | 556 | 17.3 |
| Karnali | 911 | 6.0 | 490 | 6.0 |
| Sudurpaschim | 985 | 8.6 | 546 | 8.8 |
| **Native language** | | | | |
| Nepali | 5,423 | 75.4 | 2,933 | 76.1 |
| Maithili | 793 | 13.6 | 426 | 13.4 |
| Bhojpuri | 343 | 5.6 | 188 | 5.6 |
| Other | 398 | 5.4 | 202 | 4.9 |
| **Marital status** | | | | |
| Never Married | 1,527 | 22.3 | 833 | 22.9 |
| Married or Living together | 5,205 | 74.6 | 2,790 | 73.8 |
| Widow/Divorced/Separated | 225 | 3.1 | 126 | 3.3 |
| **Women's occupation** | | | | |
| Not Working | 1,637 | 27.4 | 891 | 27.1 |
| Services | 937 | 16.1 | 513 | 16.8 |
| Agriculture-Self employed | 3,836 | 48.3 | 2,040 | 47.5 |
| Manual | 547 | 8.3 | 305 | 8.5 |
| **Women's education** | | | | |
| No education | 1,904 | 25.5 | 1,036 | 25.9 |
| Primary | 2,224 | 31.3 | 1,181 | 30.5 |
| Some secondary | 1,782 | 25.9 | 966 | 25.6 |
| SLC and above | 1,047 | 17.3 | 566 | 18.0 |
| **Women's age** | | | | |
| 15–29 yrs | 3,526 | 50.2 | 1,875 | 49.2 |
| 30–39 yrs | 1,966 | 29.1 | 1,083 | 30 |
| 40–49 yrs | 1,465 | 20.7 | 791 | 20.8 |
| **Caste/ethnicity** | | | | |
| Brahmin Hill | 655 | 10.2 | 360 | 10.1 |
| Chhetri Hill | 1,642 | 18.3 | 887 | 18.8 |
| Terai Caste | 886 | 15.6 | 462 | 15.0 |
| Dalit | 1,158 | 15.0 | 636 | 15.1 |

*(Continued)*

**Table 1.** (Continued)

| Variables | Overweight/obesity and smoking | | Hypertension | |
|---|---|---|---|---|
| | Unweighted Count | Weighted %/Mean (SD) | Unweighted Count | Weighted %/Mean (SD) |
| Hill Janajati | 1,698 | 26.3 | 913 | 26.4 |
| Terai Janajati | 697 | 10.5 | 375 | 10.6 |
| Muslim | 221 | 4.1 | 116 | 4.0 |
| **Wealth index** | | | | |
| Poorest | 1,837 | 17.3 | 993 | 17.3 |
| Poorer | 1,478 | 20.1 | 817 | 20.6 |
| Middle | 1,404 | 20.2 | 720 | 19.0 |
| Richer | 1,198 | 19.9 | 657 | 20.4 |
| Richest | 1,040 | 22.6 | 562 | 22.6 |

Note

* analytical sample size for outcome variable (overweight/obesity and smoking status)

**analytical sample size for outcome variable (Hypertension) in NDHS 202

widow/divorced/separated (36.5%), service holders (45.6%), higher educated (32.4%), hill Janajati (42.2%), the richest households (46.7%) (p<0.001). In terms of hypertension, 9.3% of women were suffering from hypertension. Women aged 40–49 years had the highest prevalence of hypertension (24.8%) (p<0.001). In addition to this, the prevalence of hypertension was higher among women who were not living with husbands or widowed (24%), with no education (14.5%), Nepalese native speakers (11%), and from hill Janajati families (13.2%) (p<0.001).

Furthermore, we assessed the prevalence of risk factors for NCD risk factors by the number of factors across different age groups (Fig 2). Over one-fourth of the participants (28.49%) had one NCD risk factors, and 6.85% showcased the prevalence of two NCD risk factors.

Table 3 depicts the adjusted association between sociodemographic characteristics and NCD risk factors derived from the modified Poisson regression models. Women's age was strongly associated with current smoking for NCDs. There was a higher prevalence of smoking among women aged 40–49 years (APR:1.77; 95% CI:1.79–5.01) and women with no education (APR: 3.06; 95% CI1.50–6.25) compared to their counterparts. The prevalence ratio of the women from Bagmati (APR:2.54; 95% CI:1.59–4.04) and Karnali (APR:1.88; 95% CI:1.30–2.73) was relatively higher than women from Sudurpaschim province. Compared to those who were not employed, those whose occupations involved agriculture or self-employment (APR:2.92; 95% CI:1.63–5.24) and manual work (APR: 3.60; 95% CI: 1.88–6.90) had a higher prevalence ratio for current smoking.

The prevalence ratio for overweight and obesity in the age 30–39 years (APR:1.80; 95% CI: 1.60–2.02) and 40–49 years (APR:2.01; 95% CI:1.77–2.29) was significantly higher than 15–29 years. Married women (APR:3.02; 95% CI:2.43–3.76) and widowed/divorced/separated women (APR:2.85; 95% CI:2.14–3.80) illustrated a higher prevalence of overweight/obesity compared to the never-married women. Overweight and obesity were almost 1.5 times higher among the women from Koshi, Madhesh, and Lumbini than the women from Sudurpaschim. Likewise, women from Bagmati (APR:1.69; 95% CI:1.40–2.04), Karnali (APR:1.28; 95% CI: 1.03–1.59), and Gandaki (APR:1.60; 95% CI:1.33–1.94) had also higher prevalence ratio for overweight/obesity compared to Sudurpaschim. Moreover, the prevalence of overweight and obesity was highest in the richest (APR:2.23; 95% CI:1.86–2.67), richer (APR:2.02; 95%

**Table 2. Prevalence of non-communicable disease risk factors among reproductive-aged women (15–49 years) (NDHS 2022).**

| Variables | Current smoking | | Overweight and obesity | | Hypertension | |
|---|---|---|---|---|---|---|
| | Number | Yes (%) (95% CI) | Number | Yes (%) (95% CI) | Number | Yes (%) (95% CI) |
| **Total** | **6,980** | **3.8 (3.3–4.4)** | **6,980** | **29.2 (27.6–30.9)** | **3,765** | **9.6 (8.4–10.8)** |
| **Place of residence** | | p<0.01 | | p<0.01 | | p = 0.377 |
| Urban | 4,791 | 3.2 (2.6–3.9) | 4,791 | 32.6 (30.5–34.9) | 2,595 | 9.9 (8.4–11.6) |
| Rural | 2,189 | 5.3 (4.4–6.3) | 2,189 | 21.7 (19.8–23.6) | 1,170 | 8.9 (7.3–10.7) |
| **Ecological zone** | | p<0.001 | | p<0.001 | | p<0.05 |
| Mountain | 356 | 8.4 (5.8–12.2) | 356 | 20.9 (14.1–29.7) | 185 | 11.0 (5.5–21.0) |
| Hill | 2,839 | 5.1 (4.1–6.2) | 2,839 | 35.8 (33.2–38.4) | 1,548 | 11.7 (9.9–13.8) |
| Terai | 3,785 | 2.5 (1.9–3.2) | 3,785 | 25.1 (22.6–27.7) | 2,032 | 7.8 (6.4–9.4) |
| **Province** | | p<0.001 | | p<0.001 | | p<0.001 |
| Koshi | 1,183 | 3.6 (2.5–5.3) | 1,183 | 28.3 (24.8–32.1) | 640 | 13.4 (10.2–17.3) |
| Madhesh | 1,356 | 0.6 (0.3–1.3) | 1,356 | 18.5 (15.2–22.2) | 743 | 5.5 (3.8–7.9) |
| Bagmati | 1,494 | 6.2 (4.7–8.3) | 1,494 | 44.3 (40.5–48.2) | 812 | 11.6 (8.7–15.3) |
| Gandaki | 667 | 2.6 (1.6–4.1) | 667 | 40.3 (36.0–44.8) | 359 | 12.0 (8.9–15.9) |
| Lumbini | 1,259 | 3.3 (2.2–5.1) | 1,259 | 28.2 (24.1–32.8) | 652 | 8.7 (6.3–11.8) |
| Karnali | 421 | 9.1 (7.2–11.5) | 421 | 18.8 (14.8–23.5) | 226 | 8.2 (5.6–11.9) |
| Sudurpaschim | 600 | 4.2 (3.0–5.9) | 600 | 14.7 (11.7–18.3) | 332 | 6.3 (4.5–8.7) |
| **Native language** | | p<0.001 | | p<0.001 | | p<0.001 |
| Nepali | 5,264 | 4.6 (4.0–5.4) | 5,264 | 33.4 (31.5–35.3) | 2,863 | 11.0 (9.6–12.6) |
| Maithili | 948 | 0.9 (0.4–1.8) | 948 | 15.0 (11.8–18.9) | 505 | 5.5 (3.6–8.4) |
| Bhojpuri | 388 | 0.1 (0.0–1.1) | 388 | 15.4 (12.2–19.3) | 210 | 2.1 (0.8–5.3) |
| Other* | 380 | 4.1 (2.3–7.2) | 380 | 20.9 (16.5–26.1) | 186 | 6.7 (4.1–10.7) |
| **Marital status** | | p<0.001 | | p<0.001 | | p<0.001 |
| Never Married | 1,556 | 1.1 (0.6–2.0) | 1,556 | 8.7(6.8–11.2) | 862 | 3.9 (2.6–6.0) |
| Married or Living together | 5,208 | 4.4 (3.8–5.2) | 5,208 | 35(33.0–37.0) | 2,779 | 10.7 (9.3–12.2) |
| Widow/Divorced/Separated | 216 | 9.6 (6.2–14.5) | 216 | 36.5(29.1–44.7) | 123 | 24.0 (16.4–33.6) |
| **Women's occupation** | | p<0.001 | | p<0.001 | | p<0.01 |
| Not Working | 1,910 | 0.9 (0.5–1.6) | 1,910 | 26.9 (24.0–30.0) | 1,020 | 7.0 (5.1–9.5) |
| Services | 1,121 | 2.3 (1.4–3.8) | 1,121 | 45.6 (41.5–49.8) | 634 | 12.1 (9.2–15.8) |
| Agriculture-Self employed | 3,371 | 5.5 (4.7–6.4) | 3,371 | 24.6 (22.7–26.5) | 1,789 | 9.3 (7.8–11.0) |
| Manual | 578 | 7.1 (4.8–10.5) | 578 | 31.9 (27.0–37.3) | 321 | 14.5 (10.5–19.7) |
| **Women's education** | | p<0.001 | | p<0.001 | | p<0.001 |
| No education | 1,779 | 9 (7.6–10.5) | 1,779 | 26.4 (23.9–29.1) | 975 | 14.5 (12.0–17.4) |
| Primary | 2,183 | 3.2 (2.3–4.4) | 2,183 | 30.5 (28.0–33.1) | 1,149 | 9.4 (7.7–11.4) |
| Some secondary | 1,809 | 1.3 (0.8–2.4) | 1,809 | 28.3 (25.5–31.2) | 965 | 7.1 (5.5–9.3) |
| SLC and above | 1,209 | 1.2 (0.6–2.4) | 1,209 | 32.4 (28.8–36.1) | 676 | 6.3 (4.2–9.3) |
| **Women's age** | | p<0.001 | | p<0.001 | | p<0.001 |
| 15–29 yrs | 3,505 | 1.4 (1.0–2.1) | 3,505 | 15.9 (14.2–17.7) | 1,853 | 2.8 (2.0–3.9) |
| 30–39 yrs | 2,029 | 3.8 (3.0–4.8) | 2,029 | 41.7 (38.8–44.6) | 1,128 | 10.1 (8.1–12.5) |
| 40–49 yrs | 1,446 | 9.7 (8.2–11.6) | 1,446 | 44.1 (40.9–47.3) | 783 | 24.8 (21.1–28.9) |
| **Caste/ethnicity** | | p<0.001 | | p<0.001 | | p<0.001 |
| Brahmin Hill | 713 | 1.0 (0.5–2.2) | 713 | 35.5 (30.8–40.6) | 381 | 9.2 (5.9–14.1) |
| Chhetri Hill | 1,276 | 5.0 (4.0–6.2) | 1,276 | 26.6 (23.2–30.3) | 707 | 9.4 (7.3–12.2) |
| Terai Caste | 1,088 | 1.0 (0.5–2.0) | 1,088 | 19.4 (16.6–22.5) | 566 | 4.7 (3.1–6.9) |
| Dalit | 1,050 | 5.5 (4.0–7.4) | 1,050 | 25.9 (22.4–29.8) | 567 | 10.5 (8.0–13.6) |
| Hill Janajati | 1,833 | 5.4 (4.2–7.0) | 1,833 | 42.2 (39.4–45.0) | 994 | 13.2 (10.6–16.3) |
| Terai Janajati | 732 | 3.9 (2.6–5.8) | 732 | 16.9 (13.4–21.0) | 400 | 8.4 (5.9–11.8) |

*(Continued)*

**Table 2.** (Continued)

| Variables | Current smoking | | Overweight and obesity | | Hypertension | |
|---|---|---|---|---|---|---|
| | Number | Yes (%) (95% CI) | Number | Yes (%) (95% CI) | Number | Yes (%) (95% CI) |
| Muslim | 287 | 0.2 (0.0–1.7) | 287 | 22.5 (16.7–29.5) | 151 | 5.5 (2.5–11.9) |
| **Wealth index** | | **p<0.001** | | **p<0.001** | | p = 0.05 |
| Poorest | 1,207 | 8.1 (6.8–9.7) | 1,207 | 15.9 (13.9–18.1) | 650 | 9.8 (7.5–12.7) |
| Poorer | 1,400 | 3.9 (3.0–5.1) | 1,400 | 20.2 (17.8–22.8) | 778 | 7.2 (5.4–9.4) |
| Middle | 1,411 | 2.6 (1.8–3.8) | 1,411 | 23.5 (21.1–26.2) | 717 | 8.0 (6.1–10.4) |
| Richer | 1,387 | 3.4 (2.3–5.0) | 1,387 | 35.8 (32.3–39.5) | 769 | 12.4 (9.9–15.5) |
| Richest | 1,574 | 2.0 (1.2–3.1) | 1,574 | 46.7 (42.8–50.6) | 852 | 10.4 (7.4–14.3) |

Note

*other native language includes other indigenous language spoken throughout the country such as Tamang, Newar, Magar, Limbu, Tharu, etc.

CI:1.71–2.37), middle (APR:1.58; 95% CI:1.34–1.87), and poorer (APR: 1.36; 95% CI:1.15–1.60) individuals relative to those belonging from poorest households.

Similarly, the prevalence of hypertension was significantly higher in the 30–39 years (APR: 3.42; 95% CI:2.21–5.27) and 40–49 years (APR: 8.47; 95% CI:5.47–13.11) age groups compared to the younger women. Hypertension was 2.2 (APR: 2.20; 95% CI:1.41–3.33) times higher among the women from Koshi than those from Sudurpaschim.

## Mean number of NCD risk factors and multivariable analysis of clustering of NCD risk factors

This study investigated the clustering of NCD risk factors in our sample, ranging from 0 to 3, within individuals. We used a multivariable Poisson regression model with robust standard error. **Table 4** presents that women aged 40–49 years and 30–39 years were, on average three folds (ARR:3.19; 95% CI:2.68–3.80) and twice (ARR: 2.24; 95% CI:1.90–2.64) likely to experience multiple NCDs risk factors relative to the younger age group of women. In comparison with the never-married women, married (ARR:1.91; 95% CI:1.49–2.45) and widowed/divorced/separated women (ARR:2.25; 95% CI:1.64–3.08) were more likely to multiple NCDs

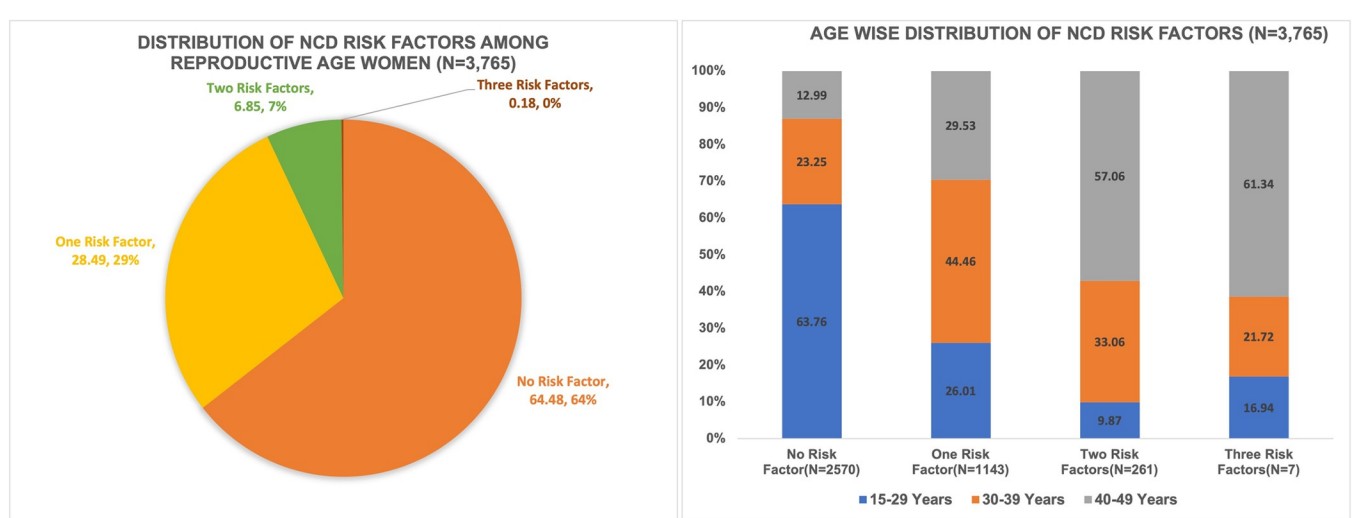

**Fig 2. Prevalence of a number of non-communicable disease risk factors among reproductive-aged women (NDHS 2022).**

**Table 3. Association between sociodemographic characteristics and non-communicable disease risk factors in NDHS 2022.**

| Variables | Current smoking, APR (95% CI) | Overweight and obesity, APR (95% CI) | Hypertension, APR (95% CI) |
|---|---|---|---|
| **Place of residence** | | | |
| Urban | 1.00 | 1.00 | 1.00 |
| Rural | 0.92 (0.72–1.17) | 1.08 (0.99–1.19) | 1.04 (0.82–1.30) |
| **Province** | | | |
| Koshi | 1.29 (0.80–2.09) | 1.52 (1.26–1.84) *** | 2.20 (1.41–3.44)*** |
| Madhesh | 0.53 (0.18–1.59) | 1.54 (1.21–1.96) *** | 1.74 (0.92–3.29) |
| Bagmati | 2.54 (1.59–4.04) *** | 1.69 (1.40–2.04) *** | 1.54 (0.95–2.51) |
| Gandaki | 0.91 (0.52–1.59) | 1.60 (1.33–1.94) *** | 1.59 (0.97–2.61) |
| Lumbini | 1.06 (0.67–1.69) | 1.50 (1.25–1.80) *** | 1.46 (0.92–2.34) |
| Karnali | 1.88 (1.30–2.73) *** | 1.28 (1.03–1.59) * | 1.28 (0.78–2.09) |
| Sudurpaschim | 1.00 | 1.00 | 1.00 |
| **Native language** | | | |
| Nepali | 13.10 (1.60–107.01) * | 1.55 (1.14–2.11) ** | 3.72 (1.30–10.65) * |
| Maithili | 5.55 (0.68–45.18) | 0.88 (0.64–1.20) | 2.34 (0.80–6.85) |
| Bhojpuri | 1.00 | 1.00 | 1.00 |
| Other | 14.47 (1.69–123.95) * | 1.37 (0.96–1.95) | 3.30 (1.01–10.81) * |
| **Marital status** | | | |
| Never Married | 1.00 | 1.00 | 1.00 |
| Married or Living together | 1.18 (0.56–2.48) | 3.02 (2.43–3.76) *** | 0.87 (0.53–1.42) |
| Widow/Divorced/Separated | 1.47 (0.64–3.34) | 2.85 (2.14–3.80) *** | 1.10 (0.60–2.02) |
| **Women's occupation** | | | |
| Not Working | 1.00 | 1.00 | 1.00 |
| Services | 2.08 (0.94–4.62) | 1.13 (1.01–1.27) * | 1.17 (0.81–1.69) |
| Agriculture-Self employed | 2.92 (1.63–5.24) *** | 0.94 (0.84–1.05) | 0.84 (0.61–1.17) |
| Manual | 3.60 (1.88–6.90) *** | 0.93 (0.79–1.08) | 1.34 (0.90–2.00) |
| **Women's education** | | | |
| No education | 3.06 (1.50–6.25) ** | 0.86 (0.73–1.00) | 1.19 (0.74–1.92) |
| Primary | 1.59 (0.76–3.34) | 1.02 (0.90–1.17) | 1.15 (0.75–1.75) |
| Some secondary | 0.93 (0.42–2.07) | 1.03 (0.91–1.17) | 1.01 (0.65–1.55) |
| SLC and above | 1.00 | 1.00 | 1.00 |
| **Women's age in years** | | | |
| 15–29 | 1.00 | 1.00 | 1.00 |
| 30–39 | 1.53 (0.92–2.54) | 1.80 (1.60–2.02) *** | 3.42 (2.21–5.27) *** |
| 40–49 | 2.99 (1.79–5.01) *** | 2.01 (1.77–2.29) *** | 8.47 (5.47–13.11) *** |
| **Caste/ethnicity** | | | |
| Brahmin Hill | 1.00 | 1.00 | 1.00 |
| Chhetri Hill | 3.13 (1.32–7.42) ** | 1.07 (0.92–1.25) | 1.16 (0.73–1.84) |
| Terai Caste | 2.53 (0.68–9.36) | 0.99 (0.81–1.20) | 0.80 (0.44–1.44) |
| Dalit | 4.22 (1.75–10.20) ** | 1.33 (1.13–1.57) *** | 1.45 (0.90–2.33) |
| Hill Janajati | 3.25 (1.38–7.68) ** | 1.44 (1.26–1.64) *** | 1.37 (0.90–2.08) |
| Terai Janajati | 3.12 (1.18–8.27) * | 0.77 (0.62–0.96) * | 1.06 (0.63–1.79) |
| Muslim | 0.45 (0.05–3.90) | 1.30 (0.98–1.73) | 0.90 (0.40–2.03) |
| **Wealth index** | | | |
| Poorest | 1.00 | 1.00 | 1.00 |
| Poorer | 0.68 (0.49–0.93)* | 1.36 (1.15–1.60)*** | 0.70 (0.49–1.00) |
| Middle | 0.54 (0.36–0.82)** | 1.58 (1.34–1.87)*** | 0.84 (0.58–1.23) |
| Richer | 0.77 (0.50–1.18) | 2.02 (1.71–2.37)*** | 1.16 (0.81–1.65) |

*(Continued)*

**Table 3.** (Continued)

| Variables | Current smoking, APR (95% CI) | Overweight and obesity, APR (95% CI) | Hypertension, APR (95% CI) |
|---|---|---|---|
| Richest | 0.57 (0.33–0.97)* | 2.23 (1.86–2.67)*** | 0.84 (0.53–1.32) |

Note: APR: Adjusted prevalence ratio

***Significant at p-value < 0.001

**Significant at p-value < 0.01

*Significant at p-value < 0.05; 1.00: refers to the chosen reference category of the explanatory variable in the model

risk factors. Individuals in Koshi (ARR:1.74; 95% CI:1.41–2.15), Bagmati (ARR:1.71; 95% CI:1.38–2.11), Gandaki (ARR:1.65; 95% CI:1.33–2.06), Lumbini (ARR:1.56, 95% CI:1.26–1.92), and Karnali provinces (ARR:1.55; 95% CI:1.24–1.94) were staying with a somewhat higher number of NCD risk factors on average compared to the individuals in Sudurpaschim province. Similarly, women belonging to the richer (ARR:1.56, 95% CI:1.31–1.84) and richest wealth quintile (ARR:1.5; 95% CI: 1.24–1.85) were more likely to have multiple risk factors of NCD on average relative to women from the poorest wealth quintile. Women who were on service work (ARR:1.23; 95% CI: 1.06–1.43) were also more likely to experience multiple NCD risk factors compared to women not on work. In contrast, women speaking Maithili (ARR:0.54; 95% CI:0.41–0.70) and Bhojpuri (ARR:0.39; 95% CI:0.26–0.59) as native languages were less likely to experience multiple NCD risk factors on average relative to Nepali native speaker women.

## Discussion

This cross-sectional study estimated the prevalence of NCD risk factors (current smoking, overweight/obesity, and hypertension) in Nepal and changes in prevalence between 2016 and 2022 (at the national and province levels), as well as investigated the association of sociodemographic factors with the prevalence of NCDs risk factors using nationally representative survey [23]. The study highlighted the high prevalence of certain NCD risk factors, such as smoking, being overweight or obesity, and hypertension, among women of reproductive age. It also showed a notable variation in the distribution of these risk factors based on sociodemographic and geographic characteristics, including changes over time at the national and provincial levels.

Our findings suggest that the prevalence of smoking, overweight/obesity, and hypertension were 3.9%, 29.2%, and 9.6%, respectively. These prevalence figures demonstrate a relatively higher prevalence of NCD risk factors among women of reproductive age. Our findings are consistent with past studies from LMICs, which have shown growth in NCD risk factors burden among women of reproductive age 15–49 years [61]. Potential underlying reasons behind the increase in NCD risk factors could be unhealthy food habits, physical inactivity, and economic growth, leading to nonessential and junk food expenses [62]. However, it's important to note that the distribution of these established NCD risk factors varies across different explanatory variables.

### Current smoking

Our study found a high prevalence of smoking among women who were aged 40+ years, illiterate, engaged in agriculture/manual work, and from poor socioeconomic status. Our findings were similar to those of previous studies in Nepal [15] and Bangladesh [12]. This decline is consistent with reports from the WHO, which estimated a sharp decline from 8.9% (2000) to 1.6% (2025) among women in the South East Asia Regional Office [63].

**Table 4. Mean number of NCD risk factors and multivariable analysis of clustering of NCD risk factors (weighted n = 3,765) NDHS 2022.**

| Variables | Mean number (SD) | Clustering of NCD risk factors, ARR (95% CI) |
|---|---|---|
| **Total** | 0.43 (0.63) | |
| **Place of residence** | | |
| Urban | 0.46 (0.64) | 1.05 (0.94–1.17) |
| Rural | 0.36 (0.60) | 1.00 |
| **Province** | | |
| Koshi | 0.45 (0.64) | 1.74 (1.41–2.15) *** |
| Madhesh | 0.25 (0.51) | 1.60 (1.19–2.13) ** |
| Bagmati | 0.61 (0.70) | 1.71 (1.38–2.11) *** |
| Gandaki | 0.56 (0.67) | 1.65 (1.33–2.06) *** |
| Lumbini | 0.41 (0.62) | 1.56 (1.26–1.92) *** |
| Karnali | 0.38 (0.59) | 1.55 (1.24–1.94) *** |
| Sudurpaschim | 0.24 (0.47) | 1.00 |
| **Native language** | | |
| Nepali | 0.49 (0.65) | 2.57 (1.69–3.90) *** |
| Maithili | 0.22 (0.47) | 1.38 (0.89–2.11) |
| Bhojpuri | 0.15 (0.36) | 1.00 |
| Other | 0.35 (0.57) | 2.62 (1.63–4.21) *** |
| **Marital status** | | |
| Never Married | 0.14 (0.41) | 1.00 |
| Married or Living together | 0.50 (0.64) | 1.91 (1.49–2.45) *** |
| Widow/Divorced/Separated | 0.80 (0.80) | 2.25 (1.64–3.08) *** |
| **Women's occupation** | | |
| Not Working | 0.33 (0.56) | 1.00 |
| Services | 0.64 (0.68) | 1.23 (1.06–1.43) ** |
| Agriculture-Self-employed | 0.39 (0.61) | 0.99 (0.86–1.14) |
| Manual | 0.52 (0.71) | 1.14 (0.94–1.38) |
| **Women's education** | | |
| No education | 0.53 (0.69) | 1.05 (0.87–1.26) |
| Primary | 0.41 (0.62) | 0.98 (0.83–1.16) |
| Some secondary | 0.35 (0.57) | 0.93 (0.79–1.09) |
| SLC and above | 0.41 (0.60) | 1.00 |
| **Women's age** | | |
| 15–29 yrs | 0.18 (0.42) | 1.00 |
| 30–39 yrs | 0.58 (0.64) | 2.24 (1.90–2.64) *** |
| 40–49 | 0.80 (0.76) | 3.19 (2.68–3.80) *** |
| **Caste/ethnicity** | | |
| Brahmin Hill | 0.45 (0.62) | 1.00 |
| Chhetri Hill | 0.41 (0.60) | 1.13 (0.93–1.36) |
| Terai Caste | 0.26 (0.51) | 1.06 (0.81–1.37) |
| Dalit | 0.41 (0.63) | 1.42 (1.15–1.75) ** |
| Hill Janajati | 0.61 (0.70) | 1.44 (1.21–1.72) *** |
| Terai Janajati | 0.28 (0.54) | 0.84 (0.64–1.09) |
| Muslim | 0.31 (0.54) | 1.24 (0.88–1.74) |
| **Wealth index** | | |
| Poorest | 0.33 (0.56) | 1.00 |
| Poorer | 0.32 (0.56) | 1.06 (0.89–1.25) |

(*Continued*)

**Table 4.** (Continued)

| Variables | Mean number (SD) | Clustering of NCD risk factors, ARR (95% CI) |
|---|---|---|
| Middle | 0.33 (0.57) | 1.15 (0.96–1.38) |
| Richer | 0.52 (0.68) | 1.56 (1.31–1.84) *** |
| Richest | 0.60 (0.68) | 1.52 (1.24–1.85) *** |

Note: ARR: Adjusted Risk Ratio, SD: Standard Deviation

***Significant at p-value < 0.001

**Significant at p-value < 0.01

*Significant at p-value < 0.05; 1.00: refers to the chosen reference category of the explanatory variable in the model

More than three-fold prevalence of smoking among those aged 40–49 years has been consistent with studies from Nepal [12, 15, 64]. Older women might have misunderstandings, such as thinking that quitting smoking later in life offers no advantages or that smoking a small number of cigarettes carries no adverse health effects [65]. The prevalence of current smoking (3.9%) has declined significantly from the previous round of NDHS (6.0%), with a steep decline in Sudurpaschim province. This decline in Nepal could be due to the implementation of a multisectoral action plan for the prevention and control of NCDs, increasing awareness among relatively younger reproductive-age women of the harmful effects of smoking, and strict enforcement of directives for the packaging of tobacco products.

## Hypertension & overweight/obesity

Over one in ten reproductive-aged women in Koshi, Gandaki, and Bagmati provinces were hypertensive in 2022. The prevalence of hypertension was significantly higher among women from the older age group and Koshi province relative to the younger age group and Sudurpaschim province, respectively. Certain factors, including high salt intake, low consumption of fruits and vegetables, physical inactivity, and obesity, as well as emerging risk factors like air pollution, urbanization, and the depletion of green spaces, may contribute to elevated hypertension [66].

The study also found a higher prevalence of overweight and obesity among women residing in Bagmati and Gandaki province, those who are married, widowed/divorced/separated, and women employed in the service sector, as well as among older women, disadvantaged Dalit and hill Janajati groups, and relatively wealthier segments of the population. The overall high prevalence of obesity might be a contributing factor behind the rise in hypertension between 2016 and 2022. Overall, 75% of the incidence of hypertension is estimated to be associated with overweight and obesity, irrespective of gender [67]. However, obesity might not be a contributing factor behind the increase in hypertension among women from Koshi province, and there might be other underlying factors that need to be further investigated. A potential explanation beyond obesity behind increasing hypertension could be the worsening air quality of Nepal over the years, as it has been established as an independent risk factor for high blood pressure [68]. The air quality of Nepal is unhealthy and 8.5 times more polluted compared to the WHO annual air quality value [69], and the situation, especially in the capital city, has worsened over the years [70].

## Clustering of NCD risk factors

Clustering of NCD risk factors appears to be more prevalent among older individuals, those from wealthier segments of the population, disadvantaged Dalits and Janajatis, as well as among married, widowed, divorced, or separated individuals, and women employed in the

service sector. We found that about 6.6% of the study participants reported having double risk factors, consistent with the earlier study, which reported 6.5% [15]. We observed that the clustering of NCD risk factors increased with age and wealth index. This finding aligns with the increasing NCD risk factors identified in previous studies from various countries, including Nepal [12, 15, 71]. The clustering of NCD risk factors among richer/richest women could be attributed to a sedentary lifestyle. This lifestyle adoption may also explain the relatively higher clustering of NCD risk factors observed across different provinces, such as Koshi, Bagmati, and Gandaki. This study showed that the likelihood of multiple NCD risk factors was higher among women in service roles. This finding aligns with previous systematic reviews on the prevalence of NCD risk factors among working women, which identified several risks such as low physical activity, sedentary lifestyles, and poor dietary habits like skipping breakfast, frequently eating junk food, and consuming few vegetables and fruits [72]. Similarly, double work burden and workplace stresses, poor welfare, and abuse by employers were NCD risk factors among working women [72]. However, working women were not found to be associated with higher odds of multiple NCD risk factors in previous studies done among women of reproductive age in Nepal [15] and Bangladesh [12].

The observed province-wise trend in risk factor clustering aligns with the findings of previous studies in Nepal [15]. Furthermore, the multidimensional poverty rate in these three provinces, as indicated by an economic survey conducted in 2018–2019, is below the national average [38]. This lower poverty rate could have facilitated urbanization and changes in dietary patterns, thereby contributing to the persistent clustering of risk factors among women in specific provinces over time. Similarly, the higher prevalence of clustered risk factors among Dalit and Hill Janajati groups might be attributed to intertwined factors such as wealth inequalities, residence in remote areas, lower educational attainment, and certain lifestyle practices [73, 74], consistent with other studies [15].

## Implications for policy and programs

The higher prevalence of smoking among older women indicates the necessity for targeted behavioural intervention aimed at addressing specific misconceptions among older female smokers. Targeted behavioural interventions could include establishing community support groups, integrating smoking cessation programs into primary healthcare, and applying mobile health interventions for older women with strong digital literacy. Moreover, effective enforcement of tobacco control policies is essential. Additionally, continuous monitoring and compliance with tobacco control policies from multiple stakeholders, along with raising taxes on tobacco products, can be cost-effective measures to reduce smoking rates among older women of reproductive age [27, 75]. Similarly, given the high number of NCD risk factors among working women, workplace interventions could be introduced to help lower these risks. Employers should prioritize their employees' health as it directly affects productivity and the quality of work [72]. Further, there is an urgent need to integrate NCD prevention, screening, diagnosis, and management into widely available maternal newborn and child health (MNCH) to prevent unfavourable maternal and newborn outcomes [76, 77]. Despite the increasing prevalence of women's NCD risk factors, access to NCD services remains limited, especially in low-resource settings, due to gender inequalities, social norms prioritizing maternal and child health needs, and poor awareness of NCDs and their risk factors, particularly in rural areas [77, 78]. The inadequacy of health system readiness further exacerbates these challenges, marked by a shortage of trained staff, guidelines, and diagnostic equipment for NCD services [33, 79]. While health system policies acknowledge the growing burden of NCDs, they fall short of effectively addressing NCD risk factors [32]. Leveraging community health

workers and volunteers in rural communities of Nepal could be a viable solution to increase awareness, early detection, and management of NCD risk factors among women in low-resource settings [80, 81]. Moreover, implementing policies and programs that specifically target women from wealthier families, Dalit and Hill Janajati communities, older women, and women in service can help prevent the clustering of NCD risk factors among reproductive-aged women (15–49 years).

## Strengths and limitations

A major strength of this study was the use of recent population-based, nationally representative data sources. The NDHS covered all seven provinces' rural and urban areas, making these findings generalizable to country and province levels. Modified Poisson regression was used for analysis to correctly estimate the APR of NCD risk factors and improve the estimates' precision of association produced by conventional logistic regression. Limitations of the study include: Firstly, since it was a cross-sectional study, the study did not establish causality. A prospective cohort study may be beneficial in determining the longitudinal progression of NCD risk factors from single to successive concurrence. Secondly, NDHS did not record other crucial clinical biomarkers (e.g., blood lipid profiles, HbA1c, serum creatinine), as well as behavioural factors (e.g., sleep duration), dietary factors (e.g., type and amount of food taken), physical activity, which are important risk factors for NCDs. So, this information could not be included in the analysis, which limits the strength of this study. Thirdly, the reliance on self-reported data for current smoking status presents another limitation. There might be the possibility of underreporting due to social stigma, particularly among younger women, which could have introduced misclassification bias into findings, leading to an underestimation of the prevalence of risk factors. Fourthly, Alcohol consumption was not included in the NCD risk factor calculation due to its very low prevalence among other ethnicities, excluding indigenous groups. Lastly, this study didn't consider the harmful use of alcohol practice among women. One of the objectives was to compare the prevalence of NCD risk factors in the last two rounds of the survey, and information about alcohol intake was missing in NDHS 2016. Future studies could address this gap by incorporating the harmful use of alcohol consumption among women of reproductive age and provide a more complete picture of NCD risk factors.

## Conclusion

Our study expands on the NCD literature in Nepal by showing the rates of risk factors rise among women of reproductive age. The prevalence of overweight/obesity was found higher in Bagmati and Gandaki province, requiring urgent attention from relevant stakeholders. NCD risk factors were highly prevalent among older women, those who are currently married, widowed, or divorced, women from the wealthiest socioeconomic groups, Dalit and Janajati castes, those with lower educational attainment, and working women. To address the growing rise of clustering of two or more risk factors, an urgent focus is needed to prioritize increasing awareness, screening, and management of NCD risk factors among women of reproductive age in communities and at the workplace. Failure to address this can increase the risk of adverse outcomes, undermining efforts to achieve the sustainable development goal of reducing premature deaths from NCD by one-third by 2030.

## Supporting information

**S1 Table. Output of independent sample proportion test (Z test) for NCD risk factors among women of reproductive age (15–49 years) between NDHS 2016 and 2022.**
(PDF)

**S1 Data.**

(DO)

## Acknowledgments

The authors would like to acknowledge the Demographic and Health Survey (DHS) program for granting access to the data of Nepal Demographic and Health Survey NDHS) 2016 and 2022.

## Author Contributions

**Conceptualization:** Barun Kumar Singh.

**Data curation:** Barun Kumar Singh.

**Formal analysis:** Barun Kumar Singh.

**Supervision:** Shiva Raj Mishra, Resham B. Khatri.

**Validation:** Shiva Raj Mishra, Resham B. Khatri.

**Writing – original draft:** Barun Kumar Singh.

**Writing – review & editing:** Barun Kumar Singh, Shiva Raj Mishra, Resham B. Khatri.

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
