## [Decision Letter · Decision Letter 0]

15 Jul 2024

PONE-D-24-21880Patterns of Non-Communicable Diseases Risk Factors Clustering Among Women of Reproductive Age in Nepal: Insights From a Nationally Representative Households SurveyPLOS ONE

Dear Dr. Singh,

Thank you for submitting your manuscript to PLOS ONE. After careful consideration, we feel that it has merit but does not fully meet PLOS ONE’s publication criteria as it currently stands. Therefore, we invite you to submit a revised version of the manuscript that addresses the points raised during the review process.

We look forward to receiving your revised manuscript.

Kind regards,

Umesh Raj Aryal, PhD

Academic Editor

PLOS ONE

Journal Requirements:

Additional Editor Comments:

Manucript needs to revise extensively

Reviewers' comments:

Reviewer's Responses to Questions

**Comments to the Author**

1. Is the manuscript technically sound, and do the data support the conclusions?

Reviewer #1: Yes

Reviewer #2: Yes

Reviewer #3: Partly

2. Has the statistical analysis been performed appropriately and rigorously? 

Reviewer #1: Yes

Reviewer #2: I Don't Know

Reviewer #3: Yes

3. Have the authors made all data underlying the findings in their manuscript fully available?

Reviewer #1: No

Reviewer #2: Yes

Reviewer #3: Yes

4. Is the manuscript presented in an intelligible fashion and written in standard English?

Reviewer #1: Yes

Reviewer #2: Yes

Reviewer #3: Yes

5. Review Comments to the Author

Reviewer #1: Title of the paper: Patterns of Non-Communicable Diseases Risk Factors Clustering Among Women of Reproductive Age in Nepal: Insights from a Nationally Representative Households Survey

Reviewer’s comment:

Title: Please avoid the last part of the title and just say “The patterns of clustering of non-communicable diseases risk factors among women of reproductive age in Nepal”.

Abstract:

Background: Please avoid using pronoun such as we, our etc. In the last sentence of the Background, avoid We aimed and say this paper aims to assess the trends of prevalence and associated factors with smoking status, overweight/obesity, and hypertension among women of reproductive age in Nepal.

Methods: Replace we examined with this paper examined…..

Keywords: please keep keywords in ascending order (a-z) without capitalization, separated by commas but not a semi colon.

Introduction: Clarify the 41 million deaths statistics by stating complete say NCDs contribute to 41 million deaths annually worldwide; 77% of these deaths occur in Low and Middle-Income Countries (LMICs). Please Provide reference?

Please choose either obese or obesity for consistency throughout the paper.

Please modify the statement the National Health Education, Information, and Communication Centre (NHEICC) conducts tobacco-specific advocacy and awareness activities to “The National Health Education, Information and communication (NHEICC) leads health promotion and behaviour change communication initiatives, including tobacco-specific advocacy and awareness programs implemented through multimedia channels, public private health care facilities, media and volunteers throughout the countries”. Add references from DOHS annual report or NHEICC reports.

Include additional reference related to Health promotion after reference 27. Such as: https://journals.sagepub.com/doi/full/10.1177/17579759221117792

https://www.nepjol.info/index.php/jhp/article/view/40957

Move the sentence to successfully fill this gap….., after your second last sentence and before the aims of this paper.

Methods

Please simplify the sentence - to identify the prevalence and significant determinants of NCD risk factors among women, this study used the NDHS 2022 data set. You can say “this study used the NDHS 2022 dataset only.

Please define your final model, what does it mean and how did you determine a final model?

Explanatory variables: What was the basis of categorization of each variable? please provide reference of each variable. If using exact categorization from NDHS 2022, please cite this report. Explain any changes made to variable categories. Please delete residence, which is duplicated. Please define each explanatory variable.

Statistical analysis: Please keep references to support your claim after the statement “Variables with a chi-square test p-value below 0.25 were included in the multivariable models”.

Start your statements with Firstly, secondly, etc, to structure your arguments.

Please insert Figure 1 and 2 in the text.

Overall comment: This paper is well written. However, please avoid repetition, ensure consistency throughout the document. Keep line numbers in the manuscript. A language review is also required.

Reviewer #2: This was an interestknf look into patterns of Non-Communicable Diseases risk factors Clustering Among Women of Reproductive Age in Nepal.

In methods: have you considered confounding factors, if so mention them

In discussion: use more literature for the comparisons and compare the findings in terms of the regions or countries you find literature from

Reviewer #3: Dear authors,

Congratulations on writing this paper. The study analysed secondary data from NDHS 2022 to determine the prevalence of three noncommunicable diseases (NCD) risk factors in Nepalese women of reproductive age group. The study further assessed the trend in the prevalence of those risk factors from 2016 to 2022. Results showed nearly one-third of those women have at least one risk factor in Nepal. The study also reported the factors associated with the prevalence and clustering of three NCD risk factors. The overall current results will have significant impact on national policy development of developing countries. The study follows the standard guidelines for reporting cross-sectional studies. The data was analysed using appropriate statistical methods. The paper has written in standard format. However, this manuscript will be further improved for clarity and accuracy by addressing the following comments.

Title

The title needs to be corrected for grammar. In the title, diseases, factors, households are nouns being used as adjectives. Therefore, these would be singular.

Is it two surveys or one survey?

Abstract

One of the objectives of the study was to assess the trends in the prevalence of risk factors. However, the results report only the prevalence rate. There are no conclusions about the trends of those risk factors.

Method

Though readers can find the sampling strategy in the cited documents, a brief paragraph is essential here as well.

A sample size of NDHS 2016 should also be stated. Similarly, please state whether the inclusion and exclusion criteria are the same for the analysed data of NDHS 2016.

Results

Table 1: The indication of * to be corrected.

Table 3: Please make the reference category consistent, either 1.o or ref…

Please report the sample size for table 4.

Data related to hypertension is different in table 1 and figure 1. Please correct or explain (N=3765 or 3749)

Figure 2 is not clear about the sample. Is it NDHS 2022 or 2016? It is not clear how sample size is 3781, when analysed hypertensive participants were 3749 for NDHS 2022 as reported in method section.

Discussion

The first paragraph seeks the important findings of the study as specified by the overall study objectives.

Data shows prevalence of clustering of risk factors was higher in women at service than those not working. Should authors need to interpret this finding?

The method and strength statement suggests that the used sample was representative of the whole country. However, the authors' conclusion is restrictive (e.g These prevalence figures demonstrate a relatively higher prevalence of NCD risk factors within the studied population). Please correct.

Does this increasing prevalence hold true “The increasing prevalence of smoking among older women”. The smoking trend in smoking between 2016 and 2022 is decreasing. The data was not analysed separately for the trend in older women. Please correct or explain.

Conclusions

The first sentence is not clear. “This study found several markers of NCD risk factors clustering among women of reproductive age in Nepal”. This sentence is to be rewritten. Based on data, women of reproductive age had a clustering of NCD risk factors.

The same comment applies to the conclusions on the trends of the risk factors.

General suggestion

Line numbers will make the review process easier to refer to the specific line.

6. PLOS authors have the option to publish the peer review history of their article (what does this mean?). If published, this will include your full peer review and any attached files.

Reviewer #1: **Yes: **Dr Shalik Ram Dhital

Reviewer #2: No

Reviewer #3: No

---

## [Author Response · Author response to Decision Letter 0]

25 Jul 2024

Dear Academic Editor

Thanks a lot for invitation to revise the manuscript. Now we have made necessary edits multiple times and have responded all the query in point by point response attached in the system.

regards, Barun kumar singh

---

## [Editor Report · Decision Letter 1]

31 Jul 2024

PONE-D-24-21880R1Patterns of Non-Communicable Disease Risk Factor Clustering Among Women of Reproductive Age in NepalPLOS ONE

Dear Dr. Singh,

Thank you for submitting your manuscript to PLOS ONE. After careful consideration, we feel that it has merit but does not fully meet PLOS ONE’s publication criteria as it currently stands. Therefore, we invite you to submit a revised version of the manuscript that addresses the points raised during the review process.

**ACADEMIC EDITOR: **Thank you addressing all comments. Please incorporate followingWith whom did the author compare hill janantis and dalit ? (see abstract)

Please incorporate the following response in the statistical analysis section for clarity to future readers.

Thanks a lot for your critical comment. As we know that, under assumption of poisson regression, mean and variance for random variable are equal, if we square the standard deviation, variance is around mean. Obviously in some cases, this assumption is violated and that was one of the reasons to choose the modified poisson regression with robust sandwich variance estimate. Moreover, mean risk factor is below 1, as this data is heavily weighted by lots of ‘zeors’ among those who do not have any NCD risk factors (approx. 64%). That’s why we didn’t interpret the mean risk factor and focused on interpreting adjusted risk ratio (ARR=i.e. ratio of count), which would be meaningful. Thank you once again for raising valid point.

We look forward to receiving your revised manuscript.

Kind regards,

Umesh Raj Aryal, PhD

Academic Editor

PLOS ONE

Journal Requirements:

Additional Editor Comments:

Thank for addressing the comments. Please go through the feedback and incorporate them.

With whom did the author compare hill janantis and dalit ? (see abstract)

Please incorporate the following response in the statistical analysis section for clarity to future readers.

Thanks a lot for your critical comment. As we know that, under assumption of poisson regression, mean and variance for random variable are equal, if we square the standard deviation, variance is around mean. Obviously in some cases, this assumption is violated and that was one of the reasons to choose the modified poisson regression with robust sandwich variance estimate. Moreover, mean risk factor is below 1, as this data is heavily weighted by lots of ‘zeors’ among those who do not have any NCD risk factors (approx. 64%). That’s why we didn’t interpret the mean risk factor and focused on interpreting adjusted risk ratio (ARR=i.e. ratio of count), which would be meaningful. Thank you once again for raising valid point.

---

## [Author Response · Author response to Decision Letter 1]

1 Aug 2024

Thanks a lot for valuable insights and feedbacks. Now we have provided by point by point response, and uploaded into the system. Once again thank you so much for your time and efforts.

regards, barun

on behalf of coauthors

---

## [Editor Report · Decision Letter 2]

9 Aug 2024

Trends and determinants of clustering for non-communicable disease risk factors in women of reproductive age in Nepal

PONE-D-24-21880R2

Dear Mr. Barun,

We’re pleased to inform you that your manuscript has been judged scientifically suitable for publication and will be formally accepted for publication once it meets all outstanding technical requirements.

Kind regards,

Umesh Raj Aryal, PhD

Academic Editor

PLOS ONE

Additional Editor Comments (optional):

No further comments
---

## [Editor Report · Acceptance letter]

24 Sep 2024

PONE-D-24-21880R2 

PLOS ONE

Dear Dr. Singh, 

I'm pleased to inform you that your manuscript has been deemed suitable for publication in PLOS ONE. Congratulations! Your manuscript is now being handed over to our production team.

Kind regards, 

on behalf of

Dr. Umesh Raj Aryal 

Academic Editor

PLOS ONE